# Proof of an Outer Membrane Target of the Efflux Inhibitor Phe-Arg-β-Naphthylamide from Random Mutagenesis

**DOI:** 10.3390/molecules24030470

**Published:** 2019-01-29

**Authors:** Sabine Schuster, Jürgen A. Bohnert, Martina Vavra, John W. Rossen, Winfried V. Kern

**Affiliations:** 1Division of Infectious Diseases, Department of Medicine II, University Hospital and Medical Center, 79106 Freiburg, Germany; martina.vavra@uniklinik-freiburg.de (M.V.); winfried.kern@uniklinik-freiburg.de (W.V.K.); 2Institute of Medical Microbiology, Greifswald University Hospital, 17475 Greifswald, Germany; juergen.bohnert@uni-greifswald.de; 3Department of Medical Microbiology and Infection Prevention, University of Groningen, University Medical Center Groningen, 9713 Groningen, The Netherlands; j.w.a.rossen@rug.nl; 4Faculty of Medicine, Albert-Ludwigs-University, 79085 Freiburg, Germany

**Keywords:** PAβN, efflux pump inhibitor, random mutagenesis, *lpxM* (*msbB*), penta-acylated lipid A, permeabilizer

## Abstract

Phe-Arg-β-naphthylamide (PAβN) has been characterized as an efflux pump inhibitor (EPI) acting on the major multidrug resistance efflux transporters of Gram-negative bacteria, such as AcrB in *Eschericha coli*. In the present study, in vitro random mutagenesis was used to evolve resistance to the sensitizing activity of PAβN with the aim of elucidating its mechanism of action. A strain was obtained that was phenotypically similar to a previously reported mutant from a serial selection approach that had no efflux-associated mutations. We could confirm that *acrB* mutations in the new mutant were unrelated to PAβN resistance. The next-generation sequencing of the two mutants revealed loss-of-function mutations in *lpxM*. An engineered *lpxM* knockout strain showed up to 16-fold decreased PAβN activity with large lipophilic drugs, while its efflux capacity, as well as the efficacy of other EPIs, remained unchanged. LpxM is responsible for the last acylation step in lipopolysaccharide (LPS) synthesis, and *lpxM* deficiency has been shown to result in penta-acylated instead of hexa-acylated lipid A. Modeling the two lipid A types revealed steric conformational changes due to underacylation. The findings provide evidence of a target site of PAβN in the LPS layer, and prove membrane activity contributing to its drug-sensitizing potency.

## 1. Introduction

Rapidly increasing rates of multidrug resistance (MDR) in aerobic and obligate anaerobic bacterial pathogens accompanied by the stagnating development of new antibiotics is one of the major public health challenges today. Alternative treatment options are urgently needed, and this may include the use of adjuvants sensitizing bacteria to antimicrobial drugs. With respect to Gram-negative bacteria, which are characterized by intrinsic resistance mechanisms due to an outer membrane (OM) barrier and MDR efflux pumps, drug potentiating agents, such as permeabilizers [1] and efflux pump inhibitors (EPIs) [2], offer a promising approach. The latter should be able to target resistance nodulation cell division (RND)-type transporters, which constitute the major MDR efflux systems in Gram-negatives. Examples are AcrAB-TolC from *Escherichia coli* and MexAB-OprM from *Pseudomonas aeruginosa*, with AcrB and MexB being the pumping components of these complexes. Most RND-type efflux pumps are characterized by an extremely broad substrate spectrum [3], and their role in the MDR of Gram-negative pathogens has been well demonstrated [4]. Agents from several chemical substance classes have been described as inhibitors of RND-type transporters [5], but none of them, to our knowledge, has reached clinical applicability so far. However, studying the mechanism(s) of action of these model EPIs could provide prospects regarding the design of new molecules. Such a model EPI is Phe-Arg-β-naphthylamide (PAβN), which is a cationic dipeptide with a naphthyl moiety and one of the first compounds reported to inhibit RND-type efflux pumps [6]. Frequently, it has been used to evaluate the efflux phenotype of Gram-negatives [7,8] and the putative substrate nature of drugs regarding RND-type transport [9]. However, from the first report of the compound, there have been indications that PAβN is not a pure EPI, and an additional mode of action by a membrane-permeabilizing activity was postulated [6,10,11,12,13]. Exploring the mode of action of PAβN was additionally complicated by the fact that the EPI itself is effluxed by RND transporters [6,14]. So far, insights of how PAβN works have predominantly come from functional studies assessing the impact on drug susceptibility and the uptake of compounds [6,10,11,12], from co-crystallization experiments with AcrB [15] and computational approaches [16,17]. Since the results predominantly suggested target sites of PAβN in the distal substrate binding pocket of AcrB, a competitive mechanism of action could be expected. However, pronounced effects of the EPI in particular with macrolides and rifamycins, which both reveal binding specificities in the proximal substrate binding pocket [18,19,20], still remained unexplained.

Recently, we reported another approach to elucidate the mechanism of action of model EPIs. By using in vitro random mutagenesis, which is also designated as directed evolution, we were able to identify target sites of 1-(1-napthyl-methyl)-piperazine (NMP) in AcrB [21]. Within the scope of that earlier study, we had obtained a mutant derived from a serial selection procedure (also referred to as in vivo mutagenesis) revealing resistance to the drug-sensitizing activity of PAβN (“PAβN-resistance”), but without any efflux-associated mutations, and with unknown reason for the observed phenotype. Here, we describe the generation of another PAβN-resistant mutant and the proof of an efflux-unrelated target of PAβN.

## 2. Results and Discussion

### 2.1. PAβN-Resistant Mutants from Random Mutagenesis Revealed Loss-of-Function Mutations in LpxM 

Previously reported PAβN-resistant mutant C5/1/17 had been derived from the sequential selection of *E. coli* 3-AG100 with increasing concentrations of clarithromycin (CLR) in the presence of PAβN [21]. Another method that was used in the same study, an in vitro random mutagenesis procedure directly targeting *acrB* by an error-prone PCR method, had failed to generate resistance to the drug-sensitizing action of the EPI. The approach had been applied to the gene regions encoding the periplasmic domain of AcrB, which is responsible for substrate recognition in RND-type transporters [22]. Since it could not be excluded that PAβN acts in an allosteric manner, we now extended in vitro random mutagenesis to the whole of AcrB. From this directed evolution approach, 4 × 10^5^ mutants were obtained and subsequently selected using a CLR–PAβN combination that inhibits the growth of the parental *E. coli* strain 3-AG100. Several macrolides including CLR are confirmed substrates of AcrB [23] with pronounced susceptibility to the action of PAβN (Figure 1a, Appendix A). PAβN was used at 25 mg/L, which is a concentration that has been demonstrated to exhibit high sensitizing potency while staying below the intrinsic MIC (minimum inhibitory concentration) in *acrB*-deficient *E. coli* mutants [24]. A single mutant, CP1, revealing stable resistance to the activity of PAβN with several drugs was achieved from the directed evolution procedure. As already seen with mutant C5/1/17, synergism in CP1 was decreased up to 16-fold with macrolides, rifamycins, and novobiocin (Figure 1a, Appendix A). Again, a significant decline in PAβN efficacy was predominantly limited to large lipophilic drugs with MW (molecular weight) >600. In contrast, marginal or no decreases in PAβN activity were detected with smaller and/or more hydrophilic agents (Figure 1a, Appendix A).

In contrast to mutant C5/1/17 and as could be expected due to the error-prone PCR method applied, *acrB* from mutant CP1 harbored four single-nucleotide mutations encoding amino acid alterations V129I, L270V, T495S, and A873V. Surprisingly, their chromosomal reconstruction in parental strain 3-AG100 did not result in any PAβN-resistance (3-AG100*acrB*_CP1_, Appendix A). Next, we substituted mutated *acrB* in mutant CP1 by wild-type *acrB*, and found the PAβN resistance phenotype maintained (CP1*acrB*_WT_, Appendix A).

Even though directed evolution approaches were used, mutations outside the targeted gene region cannot be completely excluded. Thus, we also sequenced *acrA* and *tolC* encoding the accessory proteins of AcrB. However, as with PAβN-resistant mutant C5/1/17, no mutations were detected.

Consequently, next-generation sequencing (NGS) was performed. Whole genome variant analysis revealed only one alteration shared by both mutants, C5/1/17 and CP1, in comparison to their parent 3-AG100, which was the loss-of-function of *lpxM* (*msbB*) by a frameshift and an early stop codon, respectively (Figure 1b). *LpxM* encodes an acyltransferase that is responsible for the last step in lipid A synthesis: the attachment of a secondarily bound myristic acid residue. Somerville et al. have demonstrated that the lipopolysaccharide (LPS) fatty acid pattern from *lpxM*-deficient *E. coli* mutants was lacking myristate, suggesting the occurrence of predominantly penta-acylated instead of hexa-acylated lipid A [25].

### 2.2. Proved Impact on PAβN Efficacy from an LpxM Knockout Mutant

To evaluate the impact of *lpxM* deficiency, we knocked out *lpxM* in parental strain 3-AG100. Similar to our findings with C5/1/17 and CP1, the *∆lpxM* mutant revealed significantly decreased synergistic activity of PAβN with large lipophilic drugs, whereas only marginal effects were seen with smaller and/or more hydrophilic compounds (Figure 2, Appendix A). Since Mg^2+^ ions are known to contribute to OM integrity substantially [26], and it was speculated that they might be displaced by PAβN due to its two positive charges under physiological conditions [6], we compared susceptibilities in LB (Luria/Miller broth) and in cation-adjusted medium containing higher and well-defined Ca^2+^ and Mg^2+^ concentrations. However, we could not detect any relevant impact of cation concentrations on changes in PAβN efficacy (Appendix A). Nevertheless, to avoid effects due to unphysiologically low cation availability, further assays with the aim to characterize mutant ΔlpxM were conducted in the presence of mM of MgCl_2_, and with cultures from cation-adjusted medium.

In contrast to small and rather hydrophilic drugs (considered to use predominantly porins to permeate into the bacterial cell), large and/or more hydrophobic compounds are thought to pass the OM bilayer by lipid-mediated pathways, meaning passive diffusion or self-promoted uptake [26,28]. It was also suggested that some agents, such as tetracyclines and quinolones, use both porin as well as lipid-mediated pathways [28]. This could explain the different effects on the PAβN activity that were detected with several of these antibiotics (Figure 2, Appendix A).

PAβN efficacy changes could simply be due to alterations in susceptibilities to the respective synergistic drug. However, no correlations could be found when comparing these parameters (Figure 2b). In accordance with earlier reports of *lpxM*-deficient *E. coli* mutants [25,29,30,31,32,33], the susceptibilities to most drugs were only marginally increased or unchanged with few exceptions. The fourfold decreases in the MICs of novobiocin and vancomycin (Figure 2b, Appendix A) indicated a higher OM permeability for these compounds. This might rather enhance the effectiveness of PAβN, but it was significantly decreased with novobiocin. Notably, resistance to the more hydrophilic and extremely large vancomycin was neither reducible by PAβN in wild-type *E. coli* 3-AG100 nor in the more susceptible ∆*lpxM* mutant (Figure 2b, Appendix A).

### 2.3. Hexa-Acylated Lipid A Structure from Wild-Type E. coli versus Penta-acylated from LpxM Mutants

Upon our request, the group of Wonpil Im has added the penta-acylated structure as a variant from the hexa-acylated LPS of *E. coli* K-12 to the CHARMM-GUI LPS Modeler platform [34]. Major structural changes were manifested by a significantly increased distance between the two intramolecular phosphates (Figure 3), which are supposed to play a major role in cross-linking the LPS molecules via divalent cations [10,26]. However, as already mentioned, we found a negligible impact of cation concentrations on PAβN efficacy changes in the *lpxM* mutant. Furthermore, a study with *E. coli* mutants harboring dephosphorylated lipid A revealed maintained OM integrity despite decreased cationic stabilization options [35]. In contrast to the intramolecular lipid A phosphates, the remaining fatty acid chains have moved closer together in the penta-acylated lipid A structure (Figure 3). It could be speculated that their tighter package cause altered incorporation options between adjacent LPS molecules. This might be the case for phospholipids from the inner OM leaflet, as well as for intercalating compounds from the exterior. It could also be expected that structural changes in lipid A affect the steric configuration of further LPS regions, in particular the inner core, which due to phosphates and acidic sugars could be a putative target for cationic agents such as PAβN.

### 2.4. Further Characterization of Mutant ∆lpxM

#### 2.4.1. The Activity of other EPIs

We also examined whether the drug-sensitizing efficacies of other EPIs were affected by *lpxM* deficiency. In contrast to PAβN, NMP, an arylpiperazine [36], and MBX2319, a pyranopyridine compound [37], revealed no or only subtle alteration (≤twofold, Appendix A) suggesting another mode of action and/or no membrane effects of these compounds.

#### 2.4.2. The Activity of PMBN

Polymyxin B nonapeptide (PMBN) is a large cyclic cationic peptide that is known to permeabilize the OM [1,38] by targeting the negatively charged LPS layer [28]. In contrast to PAβN, the drug-sensitizing activity of PMBN was found to be increased in the ∆*lpxM* mutant. With most of the drugs that were tested, a two- to fourfold elevated potentiating efficacy was detected, and the susceptibility to PMBN itself was enhanced by eightfold (Appendix A). Hence, different OM-compromising mechanisms of PAβN and PMBN appeared likely.

#### 2.4.3. Intracellular Dye Accumulation

To explore the functional impact of *lpxM* deficiency further, we carried out dye accumulation assays. In particular, the effects that could contribute to the decreased drug-sensitizing activity of PAβN were of interest. These included: (i) altered drug efflux capacity (ii), decreased intracellular PAβN availability, and (iii) the impaired influx of other compounds. Referring to (i), monitoring the Nile red efflux revealed similar functioning of the efflux transporters from the parental strain and mutant *lpxM* with efflux half-times of 28.3 s (±2.4, *n* = 3) and 26.3 s (±1.9, *n* = 3), respectively, whereas that of an efflux-deficient ∆*acrB* mutant was 206.7 (±6.2, *n* = 3). Notably, PAβN did not show any significant alteration in its efflux inhibitory action with this relatively small lipophilic compound (Figure 4a). Concerning (ii), the intracellular accumulation of the PAβN degradation product β-naphthylamine was slightly increased in the *lpxM* mutant, demonstrating an unimpaired influx of the EPI (Figure 4b). Disproving issue (iii) was that drug susceptibilities were not decreased (Figure 2b, Appendix A), and dye uptake was almost unchanged (Figure 4c, Table 1).

Resazurin was the only dye revealing a compromised efficacy of PAβN in mutant ∆*lpxM* (Figure 4c, Table 1), which is a finding similar to that with large lipophilic drugs. The potency of PAβN to increase the accumulation of resorufin (intracellular degradation product of resazurin) had previously been shown not only for wild-type *E. coli*, but even for efflux-deficient strains, suggesting a permeabilizing activity in addition to efflux inhibition [13]. Since the uptake of resazurin, which is a relatively small and more hydrophilic agent, was also significantly enhanced by PMBN (Table 1), an uptake pathway similar to that of large lipophilic drugs appeared likely. Further physicochemical properties, such as the polarity or stiffness of the molecular structure, supposedly determine the influx pathway, too. In contrast to resazurin, Hoechst and berberine accumulation was not increased, neither by PMBN and nor by PAβN in *E. coli* 3-AG100 and mutant ∆*lpxM* (Table 1).

## 3. Materials and Methods 

### 3.1. Bacterial Strains, Growth Conditions, and Chemicals

The *E. coli* strains and mutants that were used and generated in this study are listed in Table 2. Bacteria were grown at 37 °C overnight using cation-adjusted Müller Hinton (MH) (BBLTM Müller Hinton II, Becton Dickinson, Heidelberg, Germany) or Luria/Miller broth (LB) (Roth, Karlsruhe, Germany) as indicated. Chemicals were purchased from Sigma (Taufkirchen, Germany), NMP was purchased from Chess (Mannheim, Germany), and MBX2139 was a kind gift from Thimothy J. Opperman (Microbiotix, Worcester, MA, USA).

### 3.2. Susceptibility Testing

The MICs of drugs were determined from freshly grown overnight cultures by a standard twofold broth microdilution assay using 96-well custom plates (Merlin, Bornheim-Hersel, Germany) and inoculums of 5 × 10^5^ CFU/mL. Assays were performed in the absence and presence of the following adjuvants: PAβN, NMP, MBX2139, and PMBN at concentrations of 25 mg/L, 100 mg/L, 25 µM, and 10 mg/L, respectively. Approaches with 96-well twofold serial dilutions were also used to determine the MICs of adjuvants and dyes. MICs were determined by visual evaluation in comparison to growth-control wells.

### 3.3. In Vitro Random Mutagenesis (Directed Evolution)

Chromosome-based in vitro random mutagenesis was performed according to a procedure published previously [21]. Briefly, error-prone PCR products were achieved by amplifying *acrB* with the Mutazym II^®^ polymerase from the GeneMorph II^®^, Random Mutagenesis Kit (Stratagene, La Jolla, CA, USA). Subsequently, chromosomal wild-type a*crB* of *E. coli* 3-AG100 was substituted by the mutated PCR products using a homologous recombination method (RED/ET^®^ Counter-Selection BAC Modification Kit, Gene Bridges, Heidelberg, Germany). Resulting mutants were subjected to selection with 16 mg/L CLR in the presence of 25 mg/L PAβN. Screening for stable PAβN-resistance was done by drug susceptibility assays without and with PAβN.

### 3.4. Sequencing

Whole genome sequencing was conducted according to a protocol published earlier [40]. Raw sequencing data from mutants C5/1/17 and CP1 and from their parent 3-AG100 were deposited in ENA (European Nucleotide Archive) under sample accession number PRJEB30347. Variants were detected using the CLC Genomics Workbench v 8.0.2 (Qiagen, CLC bio, Aarhus, Denmark) by comparing with the NCBI (National Center for Biotechnology Information) reference sequence (RefSeq) NC_000913.3 from *E. coli* K-12 substr. MG1655 (accession: PRJNA57779).

Findings were confirmed by Sanger sequencing of purified PCR products (primer given in Appendix A) carried out from Microsynth SEQLAB (Göttingen, Germany).

### 3.5. Site-Directed Reconstructions

Site-directed reconstructions of *acrB* from mutant CP1 (*acrB*_CP1_) within strain 3-AG100 and vice versa of wild-type *acrB* (*acrB*_wt_) within CP1 were performed using the Counter-Selection BAC Modification Kit (Gene Bridges, Heidelberg) according to the manufacturer’s instructions. Briefly, *acrB* from the parental strain was replaced by an *rpsL-neo* cassette that was subsequently substituted by the PCR products of the respective *acrB* variant. PCR products for homologous recombination procedures were amplified using a proofreading enzyme (Q5 Hot Start High Fidelity DNA Polymerase, New England BioLabs, Frankfurt, Germany), The used oligonucleotides are shown in Appendix A. 

### 3.6. Generation of Knockout Mutants

The chromosomal *lpxM* gene knockout mutant was generated using the “Quick & Easy *E. coli* Gene Deletion” kit (Gene Bridges, Heidelberg, Germany), as indicated within the manual (oligonucleotides given in Appendix A).

### 3.7. Intracellular Dye Accumulation Assays

All of the dye accumulation assays were conducted in the presence of 1 mM of MgCl_2_, and fluorescence measurements were carried out using the fluorescence plate reader Tecan infinite M200Pro (Männedorf, Switzerland). When assays were carried out in the presence of adjuvants, PAβN and PMBN were added at concentrations of 25 mg/L and 10 mg/L, respectively.

Nile red efflux assays were performed according to a protocol published earlier [19]. Briefly, cells grown in cation-adjusted MH were deenergized with 10 µM of carbonyl cyanide 3-chlorophenylhydrazone (CCCP) and loaded with Nile red (10 µM, two hours). Efflux in washed cells was started by the addition of glucose to a final concentration of 1 mM. Fluorescence was measured at excitation and emission wavelengths of 544 nm and 650 nm, respectively.

The intracellular accumulation of PAβN was estimated by fluorescence measurement of the intracellular degradation product β-naphthylamine. Bacteria from an overnight grown cation-adjusted MH agar plate were resuspended in phosphate-buffered saline (PBS) with 0.4% D-glucose to an OD_600_ of 1. PAβN was added to a final concentration of 200 µM, and the fluorescence was measured at excitation and emission wavelengths of 320 nm and 460 nm, respectively.

Resazurin accumulation was determined by measuring the intracellular degradation product resorufin, as described previously [13]

Hoechst 33342 and berberine uptake experiments were carried out as described for PAβN accumulation. Final dye concentrations used were 2.5 µM for Hoechst and 30 mg/L for berberine. The fluorescence of accumulated Hoechst was monitored at excitation and emission wavelengths of 350 nm and 461 nm, respectively (berberine, 355 nm and 517 nm).

### 3.8. LPS Modeling and Visualization

LPS structures were modeled using the LPS Modeler provided by the CHARMM platform (http://www.charmm-gui.org) [34] with LPS from *E. coli* K-12, lipid A type 1, and *E. coli* K-12 lipid A type 3. Visualization and distance measurements were performed using the PyMOL Molecular Graphics System, Version 2.0 Schrödinger, LLC.

### 3.9. Statistical Analysis

SD values were calculated from the mean of n experiments (*n* ≥ 3, if not otherwise indicated; *n* > 3 if the SD calculated from three replicates was >10%). Statistical significance of differences was analyzed by a two-tailed *t*-tests using the software GraphPad Prism version 7.05 (*p*-values < 0.05 represent significance).

## 4. Conclusions

For the first time, the molecular basis of an OM target of PAβN was identified. The study revealed that *lpxM* deficiency, that was known to cause penta-acylated lipid A, was able to substantially decrease the sensitizing activity of PAβN with large lipophilic drugs, which are suggested to use a lipid pathway to access the bacterial cell. Our results provide the basis for further research, particularly with respect to the development of OM permeabilizers, and identify PAβN as a perfect template for the design of drug sensitizers with a dual mode of action but as an inadequate tool for the evaluation of efflux phenomena. 

## Figures and Tables

**Figure 1 molecules-24-00470-f001:**
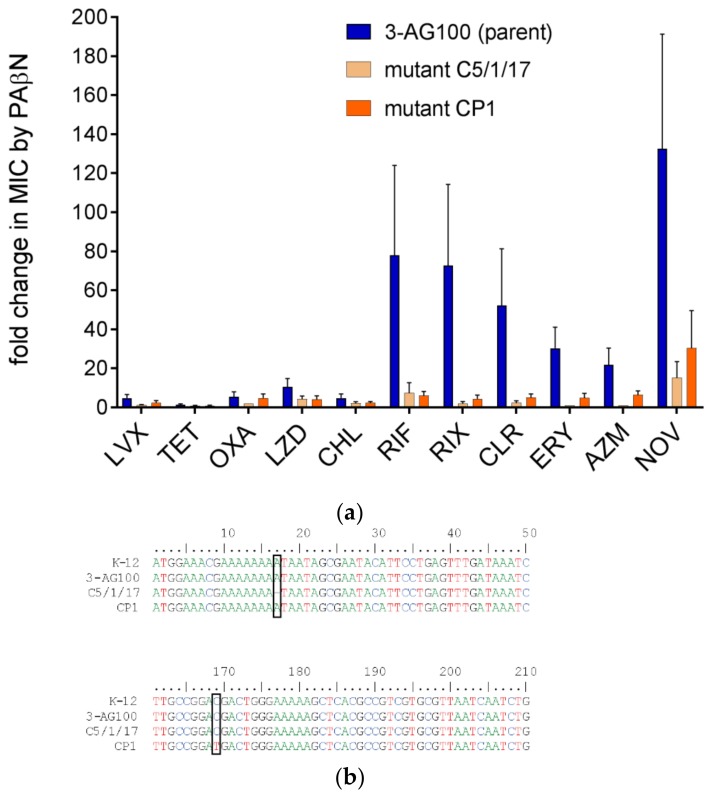
Phe-Arg-β-naphthylamide (PAβN)-resistant mutants versus parental *E. coli* 3-AG100 (**a**) PAβN efficacies with drugs shown as ratios of minimum inhibitory concentrations (MICs) without and with 25 mg/L PAβN; LVX, levofloxacin; TET, tetracycline; OXA, oxacillin; LZD, linezolid; CHL, chloramphenicol; RIF, rifampin; RIX, rifaximin; CLR, clarithromycin; ERY, erythromycin; AZM, azithromycin; NOV, novobiocin. Statistical significance determined for RIF, RIX, CLR, ERY, AZM, and NOV (mutants vs. parent 3-AG100), *p*-values ≤ 0.001 (*n* = 4); MICs are shown in Appendix A. (**b**) Nucleotide region 1–50 and 161–210 of gene *lpxM* (reference sequence *E. coli* K-12, RefSeq NC_000913.3).

**Figure 2 molecules-24-00470-f002:**
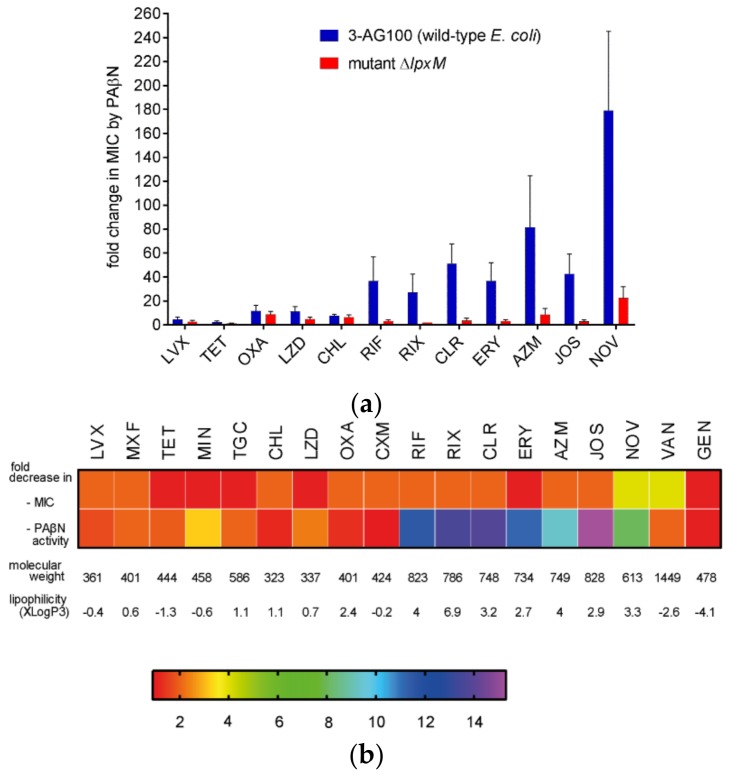
PAβN activity in mutant ∆*lpxM* in comparison to its parent 3-AG100. (**a**) Ratios of MICs without and with 25 mg/L of PAβN; Statistical significance (∆*lpxM* mutant vs. parent 3-AG100), *p*-values < 0.0001 for RIF, RIX, CLR, ERY, AZM, and NOV; other *p*-values < 0.01, except for LVX (*p*-value 0.02), and for OXA and CHL (*p*-value > 0.1; *n* ≥ 6). (**b**) Heat map depicting decreases in MIC (MIC_3-AG100_/MIC_∆lpxM_) and in PAβN activity ((MIC_3-AG100_/MIC_3-AG100+PAβN_)/(MIC_∆lpxM_/MIC_∆lpxM+PAβN_)), color scale of the heat map represents x-fold changes; molecular weights and computed partition coefficients XlogP3 of drugs from PubChem [27]. LVX, levofloxacin; MXF moxifloxacin; TET, tetracycline; MIN, minocycline; TGC, tigecycline; CHL, chloramphenicol; LZD, linezolid; OXA, oxacillin; CXM, cefuroxime; RIF, rifampin; RIX, rifaximin; CLR, clarithromycin; ERY, erythromycin; AZM, azithromycin; JOS, josamycin; NOV, novobiocin; VAN, vancomycin; GEN, gentamicin. MICs are provided in Appendix A.

**Figure 3 molecules-24-00470-f003:**
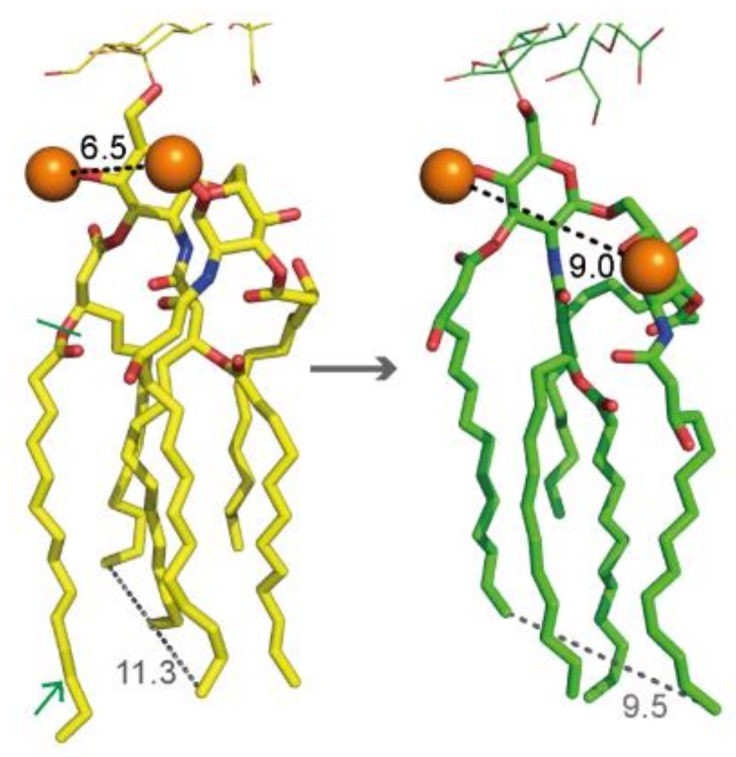
Structures of hexa-acylated and penta-acylated lipid A depicted as yellow and green sticks, respectively (red: oxygen, blue: nitrogen). Phosphates are shown as orange spheres, and distance measurements are shown as dashed lines. The green arrow indicates the myristic acid residue of hexa-acylated lipid A, and the green bar indicates its binding site in lipid A.

**Figure 4 molecules-24-00470-f004:**
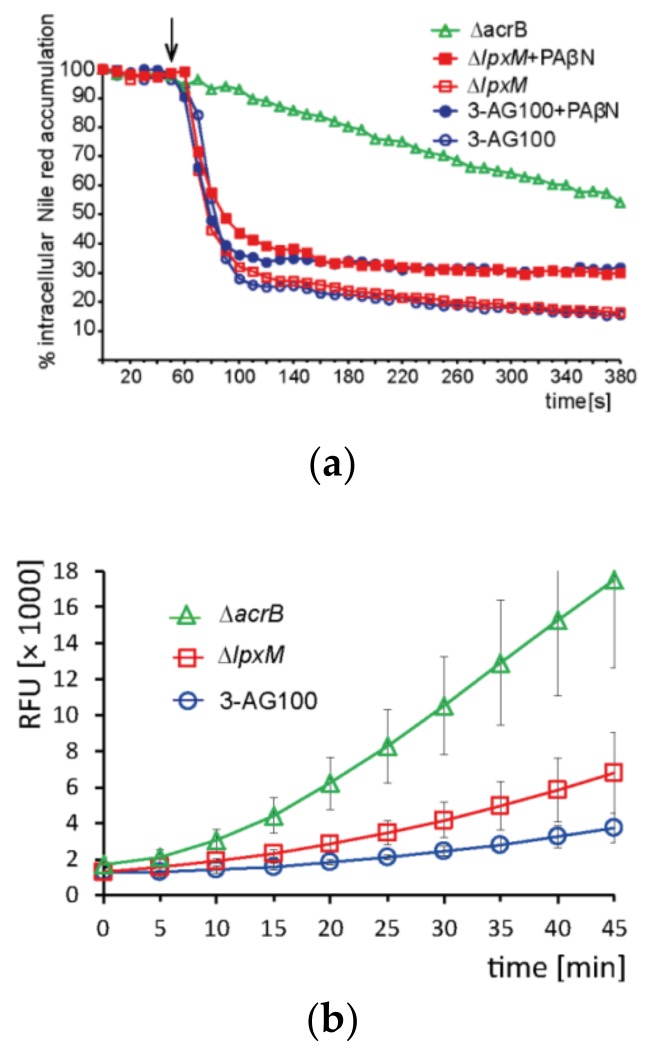
The kinetics of intracellular dye accumulation in mutant ∆*lpxM,* parental *E. coli* 3-AG100, and an efflux-deficient ∆*acrB* derivative from 3-AG100. RFU, relative fluorescence units; error bars indicate ± standard deviation (SD) from the mean. (**a**) Real-time efflux of Nile red. The arrow indicates the initiation of efflux by the addition of glucose to de-energized cells; (**b**) β-naphthylamine accumulation (degradation product of PAβN). Statistical significance (mutants vs. 3-AG100 at 45 min), *p*-values < 0.0001 (*n* = 8); (**c**), Resazurin accumulation, determined by measurement of the intracellular degradation product resorufin. Statistical significance (3-AG100 with PAβN vs. ∆*lpxM* with PAβN), *p*-value < 0.0001 (*n* = 7).

**Table 1 molecules-24-00470-t001:** Intracellular dye accumulation in the absence and presence of PAβN and polymyxin B nonapeptide (PMBN).

	% Dye Accumulation (Relative to ∆acrB Mutant without Adjuvants) ^1^								
	3-AG100			∆*lpxM*			∆*acrB*		
Dye		**+PAβN**	**+PMBN**		**+PAβN**	**+PMBN**		**+PAβN**	**+PMBN**
Resazurin	33.4 (±2)	208.4 (±11)	106.8 (±15)	33.1 (±1)	139.6 (±18)	78.1 (±9)	100 (±9)	209.7 (±4)	226.1 (±4)
Hoechst	43.1 (±6)	44.5 (±7)	44.4 (±1)	47.2 (±12)	47.6 (±11)	45.3 (±9)	100 (±11)	108.6 (±8)	108.3 (±3)
Berberine	31.5 (±7)	32.6 (±8)	18.7 (±0)	22.2 (±1)	22.3 (±1)	19.6 (±1)	100 (±14)	117.8 (±14)	90.9 (±1)

^1^ PAβN used at 25 mg/L, PMBN used at 10 mg/L. Values determined after 30 min of accumulation; SD of the mean given in parenthesis. Statistical significance proved for resazurin accumulation in the presence of PAβN and of PMBN vs. the absence of adjuvants in all strains, and for berberine accumulation with mutant ∆acrB (in the presence of PAβN vs. accumulation without adjuvants); *p*-values < 0.01; *n* = 7, assays with PMBN, *n* ≥ 2.

**Table 2 molecules-24-00470-t002:** Strains and mutants used and engineered in the present study.

*E. coli* Strains and Mutants	Description	Source
3-AG100	*E. coli* K-12 AG100 derivative; overexpression of *acrB*.	Kern et al., 2000 [39]
C5/1/17	PAβN-resistant serial selection mutant from 3-AG100.	Schuster et al., 2014 [21]
CP1	PAβN-resistant in vitro random mutagenesis mutant from 3-AG100.	This study
3-AG100*acrB*_CP1_	Site-directed mutagenesis mutant from 3-AG100.	This study
CP1*acrB*_WT_	PAβN-resistant site-directed mutagenesis mutant from CP1.	This study
∆*lpxM* ^1^	PAβN-resistant *lpxM* knockout mutant from 3-AG100.	This study
∆*acrB* ^2^	Efflux-deficient *acrB* knockout mutant from 3-AG100.	Schuster et al., 2014 [21]

^1^ Insertion of a PGK-*gb2-neo* cassette in *lpxM*; ^2^
*acrB* replaced by an *rpsL*-*neo*-cassette.

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
