# Peer review of "Proof of an Outer Membrane Target of the Efflux Inhibitor Phe-Arg-β-Naphthylamide from Random Mutagenesis"

_molecules, 2019, doi:10.3390/molecules24030470_

Round 1

Reviewer 1 Report

The MS is very interesting and important. We really need more information on the mechanisms of action of antibiotic efflux inhibitors and drug potentiating agents. The paper could be divided in two parts: 1) derivation and structural properties of LpxM mutant LPS and 2) determination of the cell envelope permeability of E. coIi LpxM mutant. I have no remarks to the first part of the study. However, the second part of the paper needs improvement. First of all, there are too many prepositions. Statement that „different OM compromising  mechanisms of PAbetaN and PMBN appeared likely“ does not explain too much. Division of antibiotics and other indicatory compounds into two groups – „large lipophilic and small and/or more hydrophilic“, idea about the mechanisms of self-promoted uptake of „large“ antibiotics should be more clearly explained. Which of the tested compounds can get into the cells using self-promoted uptake? Results presented in the chapter „Intracellular dye accumulation“ should be more clearly discussed. It is not clear were degradation of PAbetaN and resazurin occurs – in periplasm or cytosol? Why the permeability of the OM was not increased using simple procedure - Tris/EDTA treatment?

Author Response

Response to Reviewer 1

1.      Too many prepositions. In the revised manuscript, several prepositions were removed in section 2.4.3.

2.      Statement “different OM compromising mechanisms of PAbetaN and PMBN appeared likely”. In fact, the conclusion was not clear, because “to” was missing in line 179 of the original manuscript (MS) that we now included. It should be “In contrast to PAbetaN…..” We just concluded the statement from observing different impact of lpxM deficiency on the synergizing activities of PAbetaN on the one hand and PMBN on the other hand (decreased versus increased drug sensitizing activity, respectively) and from significantly increased susceptibility to PMBN itself (original MS, line 181, table S2). The latter could not be observed with PAbetaN (MICs given in Table S2).

3.      Division of antibiotics and other indicatory compounds into two groups – „large lipophilic and small and/or more hydrophilic“, idea about the mechanisms of self-promoted uptake. We had written some sentences regarding uptake mechanisms in line 126-131 (original MS). In fact, for large and/or lipophilic agents, it has been postulated that they enter the Gram-negative bacterial cell by slow diffusion through the lipid bilayer. The detailed mechanism how this could take place via the strongly hydrophilic and anionic polysaccharide layer of the LPS has still remained unclear. With respect to drugs, self-promoted uptake has been postulated for polymyxins and aminoglycosides (Nikaido, 2003). However, it appears definite that large and/or lipophilic compounds cannot pass porins which are known to be water-filled and have relatively small pore diameters (see cited literature, Nikaido 2003 and Delcour 2009). Delcour defined two pathway types, the porin-mediated and the lipid-mediated pathway (might be diffusion or self-promoted uptake). We now indicated this division more clearly in section 2.2. (line 135, “track changes” version of revised MS).

4.     Results presented in the chapter „Intracellular dye accumulation“ should be more clearly discussed. We now included additional remarks in line 218-221 (“track changes” version of revised MS) concerning our conclusions from the resazurin accumulation assays.

5.     Cleavage of PAbetaN and resazurin in cytosol or periplasm. Cleavage of amino acid naphthylamide derivatives have been reported to occur inside the cell (Lomovskaya 2001), similarly resazurin (Vidal-Aroca et al, 2009). Actually, it was not important for our question whether the agents were cleaved in the cytosol or in the periplasm. We examined, whether these compounds can cross the OM and also whether they were effluxed by AcrB (comparison with AcrB deficient mutant) by measuring the intracellular cleavage products. AcrB has been reported to extrude agents from the periplasm and also from the cytosol. We agree, without definite knowledge about where the intracellular degradation occurs (cytosol or periplasm), we cannot give a statement regarding the permeability of the inner membrane (for PAbetaN and resazurin). However, an evaluation with respect to the OM was possible (lpxM mutations target the OM).

6.     Why the permeability of the OM was not increased using simple procedure - Tris/EDTA treatment? We intended to explore the drug sensitizing activity of PAbetaN in untreated cells of wild-type and deltaLpxM strains using almost physiological conditions (cation-adjusted medium without any detergents).

Reviewer 2 Report

This is a concise and clear report of identification of lpxM as integral to the action of PABN as an efflux inhibitor of larger substrates of RND efflux systems. This is a nice piece of work which has been carefully done. I think this will be of broad interest to those studying efflux or antibiotic entry into cells. My only suggestion for improvement would be to add data for some of the smaller compounds to Figure 1 (as in Figure 2) which helps show the distinction between classes of substrates.

Author Response

Response to Reviewer 2

Figure 1 a. As recommended, we replaced Figure 1 (a) by a figure which includes data for some smaller drug compounds.

Reviewer 3 Report

The manuscript “

The manuscript “Proof of an Outer Membrane Target of the Efflux Inhibitor Phe-Arg-β-naphthylamide from Random Mutagenesis” touches a very interesting and important topic, namely the efflux-mediated drug resistance of Gram-negative bacteria, and the elucidation of the exact mechanism of action of PABN, a known EPI compound. The paper reports on research using current methods and it is written in a concise and logical fashion.

However, there are some minor revisions required before the paper could be considered for publication. I have listed these below:

The abbreviations in the manuscript should be used consistently, i.e., the first time they should be explained and later only the abbreviations should be used. In addition, a “List of abbreviations” section should be added.

Page 1, line 35: complement this sentence

Rapidly increasing rates of multi-drug resistance (MDR) in aerobic and obligate anaerobic bacterial pathogens accompanied by stagnating development of new antibiotics is one of the major public health challenges today.

I suggest citing these relevant articles:

1. New Roads Leading to Old Destinations: Efflux Pumps as Targets to Reverse Multidrug Resistance in Bacteria. Molecules 2017, 22(3), 468; https://doi.org/10.3390/molecules22030468

2. Identification and Antimicrobial Susceptibility Testing of Anaerobic Bacteria: Rubik’s Cube of Clinical Microbiology? Antibiotics 2017, 6(4), 25; https://doi.org/10.3390/antibiotics6040025

Page 1, line 37: change the word order in the sentence

“…may include the use of adjuvants sensitizing bacteria to antimicrobial drugs.”

Page 1, line 39: change the word order in the sentence

“…resistance mechanisms both due to an outer membrane (OM)…”

Page 1, line 40: complement the sentence

“…and efflux pump inhibitors (EPIs) offer a promising approach.” I suggest

Page 2, line 45: complement the sentence

“…and their role in the MDR of Gram-negative pathogens has been…”

Page 2, line 49: remove “being” from the sentence

Page 2, line 53: remove comma after EPI

Page 2, line 54: “postulated” instead of “supposed”

Page 2, line 56: “came” instead of “come”

Page 2, line 58: remove comma after AcrB

Page 2, line 62: change the word order in the sentence

“still remained unexplained”

Page 2, line 64 and 67: The use of in vitro and in vivo should by without the hyphen and in italics. This formatting should be used throughout the manuscript.

Page 2, line 74: change the word order in the sentence

“…increasing concentrations of clarithromycin (CLR)…”

Page 2, line 78: “which is” instead of “known to be”

Page 2, line 79: “Because” instead of “Since”

Page 2, line 80: “to the whole of AcrB”

Page 2, line 83: “activity” instead of “action”

Page 3, line 103: “Even though directed evolution approaches were used, mutations outside the targeted gene region cannot be completely excluded”.

Page 3, line 108: “which was the loss-of-function”

Page 3, line 110: “Sommerville et al.” et al. in italics.

Page 4, line 119: “contribute to OM integrity substantially”

Page 4, line 122: “concentrations” instead of “amounts”, “relevant” instead of “crucial”

Page 4, line 122: “changes PABN efficacy”

Page 4, line 125: remove unnecessary dot in sentence

Page 4, line 126: “which are considered to use predominantly porins to permeate into bacterial cell” should be put in brackets. “by self-promoted uptake”

Page 4, line 130: “the more or less”

Page 4, line 133: “synergistic” instead of “synergizing”

Page 4, line 136: “The 4-fold decrease in the MIC of…”

Page 4, line 137: remove the however

Page 5, line 168: “red-oxygen, blue-nitrogen”

Page 5, line 171: remove unnecessary dot

Page 5, line 179: “In contrast to…”, “was found to be increased…”

Page 7, line 221: correct the format for Celsius degree

Page 8, line 245: remove unnecessary comma after CP1

Page 8, line 267: “when” instead of “if”

Section 3.2. How did the authors evaluate the MICs of drugs? With the naked eye or with the help of a spectrophotometer? Please specify!

Section 3.7. How did the authors decide on the number (n) of experiments for each individual experiment? Please specify!

Please check if all references are according to the Journal format.

Author Response

Response to Reviewer 3

1.      List of abbreviations. To our knowledge, a list of abbreviations has not been scheduled in this journal format. We could not find an appropriate section in the manuscript template. However, we checked the consistent use of abbreviations in the manuscript and removed “MDR” from the abstract (line 18).

2.      Recommended citations. We added the citation “New Roads Leading to Old Destinations: Efflux Pumps as Targets to Reverse Multidrug Resistance in Bacteria. Molecules 2017, 22(3), 468” ( line 41, revised manuscript, “track change” version) but we did not include recommended publication 2, because MIC testing of E. coli strains was not carried out under anaerobic conditions (E. coli is a facultative anaerobic microbe and was usually tested under aerobic conditions).

3.      Word orders, replacements and sentence complementations. We included most recommended changes of word orders, word replacements, complementation of sentences, formatting and we removed unnecessary commas and dots, see the revised manuscript.

4.      MIC determination method. Section 3.2. We specified the MIC determination method by including the sentence “MICs were determined by visual evaluation in comparison to growth-control wells.”

5.      Number of experiments. Section 3.7. (Statistical analysis); the section number was wrong, we changed it to 3.9. We specified the number of experiments n (n ≥ 3, if not otherwise indicated; n > 3, if SD calculated from 3 replicates was > 10 %) in line 303-304 (revised manuscript, “track changes version”).

6.      References format.We checked the format of the references.